# Microgrids Resiliency Enhancement against Natural Catastrophes Based Multiple Cooperation of Water and Energy Hubs

**Sattar Shojaeiyan [1], Moslem Dehghani [2] and Pierluigi Siano [3,4,*]**

1    Department of Electrical Engineering, Marvdasht Branch, Islamic Azad University, Marvdasht 7371113119, Iran; shojaeiyan.mau@gmail.com
2    Department of Electrical and Electronics Engineering, Shiraz University of Technology, Shiraz 7155713876, Iran; mo.dehghani@sutech.ac.ir
3    Department of Management and Innovation Systems, University of Salerno Via Giovanni Paolo II, 132, 84084 Fisciano, Italy
4    Department of Electrical and Electronic Engineering Science, University of Johannesburg, Johannesburg 2006, South Africa
*    Correspondence: psiano@unisa.it

**Abstract:** With the ever-growing frequency of natural catastrophe occurrences such as hurricanes, floods, earthquakes, etc., the idea of resilient microgrids (MGs) has attracted more attention than before. Providing the opportunity for a multi-carrier energy supply after a natural catastrophe can lessen power losses and improve power resiliency and reliability. Critical loads within the MG can be prioritized and restored in the shortest possible time based on the condition of the network after the damaging occurrence by considering the energy hub (EH) systems and the optimum design and allocation of these multi-carrier systems. To this end, this paper aims to address the resilience framework in MGs considering sets of water and EHs (WEHs) consisting of CHP (combined heat and power), a boiler, energy storage, and a desalination unit. This study focused on considering an effective resilient scheme to restore critical loads in a short period after a natural catastrophe when the MG experiences an unpredictable event. By applying the idea of WEHs, there would be a chance of restoring the system by using two sets of WEH systems in the appropriate islanded points to restore the system and critical loads of electricity, heat, and water. For this purpose, different scenarios were considered for assessing the resiliency of the system against a natural catastrophic event that causes serious damage to the network by analyzing the energy-not-supplied (ENS) factor. Moreover, the allocated WEHs can adequately supply the electrical, water, and thermal demand loads throughout the day after the natural catastrophe. To mitigate the unforeseen variations in the renewable sources, a battery is located in the WEH, which can attend to the optimal scheduling effectively. A scenario-based method is also introduced to improve the resiliency of MGs in an uncertain environment such as electrical, heat, and water stochastic demands. The appropriate efficiency of the offered model was considered on a modified IEEE test system.

**Keywords:** resilience; water and energy hub; scenario-based method; microgrid; electricity distribution systems

## 1. Introduction

Storms, earthquakes, floods, and other weather occurrences and phenomena have the potential to impact a large number of consumers and, as a result, inflict significant harm to end-users. The imposition of substantial damages and blackouts at the distribution network level has been caused by the occurrence of adverse weather conditions and natural catastrophes, and the frequency and intensity of these incidents have been growing in recent years. Natural catastrophes have become more common in recent years as a result of

climate change and global warming [1]. Some of these instances are more likely to occur depending on a region's geographical position, while others are less probable, but the rise in their incidence based on extant documentation is an obvious truth [2]. Electrical power distribution systems are among the systems that are prone to turbulence owing to their complex structure and the diversity of technology and human aspects, and have always faced the issue of system stability, preserving dependability, and enhancing system resilience. Electrical power distribution grids are the most commonly utilized component of the electrical power grid; in reality, this section serves as a link between consumers, transmission, and production and is particularly vulnerable owing to its proximity to customers. Any inaccuracies in this part result in irreversible consumer losses. Distribution networks, being the last power grid link with end-customers, are fully susceptible, and distribution network defects account for a major portion of consumer failures. The long routes to the customers, damaged or blocked roads, and the lack of appropriate equipment in the network are the main factors that prolong equipment repair time and the recovery of lost loads during a distribution network fault. In this regard, using appropriate methods can prevent damage caused by long-term blackouts and thus improve network resilience [3].

Microgrids (MGs), as a local intelligent distribution system with readily available and secure resources, may now prepare the network for quick reaction to natural and unforeseen occurrences, therefore improving the power grid's resilience [4]. However, in addition to normal worries regarding MG project costs, these MGs may have long-term problems when confronted with unanticipated disruptions caused by human mistakes or natural calamities. The MG resilience and the ability to restore loads after a disruption is a critical problem that should be carefully studied.

The goal of resilience studies is the reaction to crucial network loads in the shortest amount of time after a natural catastrophe such as storms, floods, earthquakes, and so on. The usage of multicarrier energy systems (MCESs) known as energy hubs (EHs) is one of the most successful strategies for speedy reaction following a natural catastrophe [5]. In recent years, extensive research and investigations have brought a new notion to the subject of energy trading among different sectors of the network. The EH is one of the newest options for reducing energy usage, which may be utilized to boost efficiency, reliability, and network resilience. Rather than concentrating on a single energy carrier (such as electricity), multicarrier energy (MCE) (such as electricity, gas, and local heating) is employed as the input in this method. The EH is an MCES made up of numerous transducers, input, and output parts that mostly rely on gas and electrical infrastructure. Converters and energy systems, such as combined heat and power (CHP), transformers, gas furnaces, and storage, are all found within the hub [6]. In the event of a natural catastrophe in the distribution grid, it is feasible to convert and combine these energy resources in the most efficient manner possible to satisfy the demand for the required load on the consumer side [7]. Because the EH system's output includes electricity, and other types of energy such as heat, water, and so on, these systems should be able to balance production and consumption across all sectors to the greatest extent feasible [8]. This balancing of various energy types in EH also improves the system's resilience because, in addition to meeting electricity demand, and loads of various energies are supplied repeatedly in the distribution network, such as thermal and water loads in those specific conditions, and, as a result, the distribution network's resilience improves and system blackouts are reduced [9].

In the case of an unforeseen catastrophe, having a resilient network requires a safe operation. Due to the unpredictability of natural catastrophes, as well as the low possibility of these occurrences occurring, reliability studies and criteria cannot ensure that a system would respond adequately to these situations [10]. The capacity of a system to withstand occurrences with a low chance of occurrence but a large destructive impact is referred to as resilience. This ensures that blackouts are minimized and that regular power system functioning is resumed. Unexpected occurrences such as floods, hurricanes, earthquakes, and other natural disasters have increased the number of widespread faults in recent decades. Given our current reliance on electrical energy, energy experts and consumers are

focusing their efforts on developing a stable and cost-effective energy source. A review of resilience definitions was undertaken in [11] to clarify the idea of power system resilience. A resilience operation method for the formation of several MGs after a natural disaster in the distribution network has been studied in [12], and after the maximum critical load was retrieved by distributed generation sources following the accident, this reference only investigated the effect of distributed generation resources and MG formation on improving resilience and, in the case of natural disasters, on electricity and gas infrastructures. Other forms of energy such as heat, water, etc., are not mentioned, although in the event of natural disasters such as storms, damage to all infrastructures is possible [13]. Such events could result in the outage of important distribution network equipment and cause irreversible damage to the electrical network and must be addressed.

A self-healing technique for recovering the maximum cutoff load is presented in [14], which separates the disconnected region of the distribution network into numerous MGs powered by renewable energy sources (RESs).

In [15–17], the MG programming model is proposed to investigate the amount of resilience and expenses. The network's resilience level is then determined based on the allowable load losses in the network, ensuring that the network is resilient against the worst event or event. Finally, based on the maximum load in the network as well as the predetermined resilience level, the maximum permitted load losses can be determined, and the level of resilience can be increased based on the allowable load losses in the network, which can be determined for this work. Energy distribution resources are changed using the proposed optimization technique so that the network is resistant to the worst occurrences, load losses are decreased, and resilience is increased. The disadvantage of these references is the lack of rapid load recovery; in fact, the time required for network recovery is not discussed in this reference, although network resilience and speed are important issues that should be considered in the target function, because network resilience is a function of time.

Preventive techniques for the efficient functioning of MCE-MGs are described in [18] to guard against probable natural catastrophes or human mistakes. In [19], a preventative strategy for system resilience is developed, to improve the storm preparedness of power systems and EH networks. A preventive operating plan for dealing with extreme floods is presented in [20]. The demand response (DR) was evaluated in critical circumstances employing situational awareness in [21]. It also discusses how to adjust climate-influenced power systems using an integrated corrective preventative approach. Additionally, some of the main references related to this work are provided in Table 1.

**Table 1.** Comparison between studies related to the considered model.

| Reference | Resiliency | EH | WEH | MG | Allocation |
|---|---|---|---|---|---|
| [9] | ✗ | ✓ | ✗ | ✓ | ✗ |
| [10] | ✓ | ✓ | ✗ | ✗ | ✗ |
| [17,18] | ✓ | ✓ | ✗ | ✓ | ✗ |
| [22] | ✗ | ✗ | ✓ | ✓ | ✗ |
| [23] | ✓ | ✓ | ✗ | ✗ | ✗ |
| [24] | ✗ | ✓ | ✓ | ✗ | ✓ |
| [25] | ✓ | ✓ | ✗ | ✗ | ✗ |
| [26] | ✓ | ✓ | ✗ | ✗ | ✗ |
| Proposed Model | ✓ | ✓ | ✓ | ✓ | ✓ |

Power transmission overhead lines and underground gas supply lines were created concurrently in [27,28] to increase resilience and lessen vulnerability to major natural disasters and malicious interruption. The MG demand, which includes electric and thermal charges, is satisfied by interdependent electricity and natural gas sources in all of these sources. In these references, the suggested approach is applied to boost productivity and lower network operating costs. The freshwater production unit, whose purpose is to supply part of the water consumers' usage, is not utilized by the EH used by these

authorities. However, since drinking water contamination is one of the key concerns during occurrences such as storms, the desalination unit should be incorporated in the EH analysis. Furthermore, following a natural disaster, the authorities do not debate the recovery of loads or the balance between production and consumption. After carefully reviewing the related literature, we came up with the idea of water and energy hub (WEH) for resiliency improvement of the networked MGs. For this purpose, different scenarios were considered for assessing the resiliency of the network against a natural catastrophic event that causes serious damage to the network by analyzing the energy-not-supplied (ENS) factor. Since the MGs are incorporated with RESs, the scenario-based method is presented to capture the uncertainty associated with the electrical, heat, and water demands and electricity, natural gas, and water prices. In this work, we assumed that the heat and water system outage is caused by the natural catastrophe, and the heat and water consumption of the MG buses is supplied only by the WEHs. This concept has not been explored in the related literature. Additionally, it should be mentioned that the studied 24-bus networked MGs case was chosen to perform simulations on it and demonstrate the effectiveness of the work. This approach applies to other networked MGs that have RESs and can host WEHs. Therefore, the innovations of this study are briefed as follows:

- Developing an effective structure of a multi-WEH system to improve the resiliency of the MGs against natural catastrophes;
- Representing a scenario-based approach to demonstrate the critical demands of various energy restorations in less potential time using the WEH;
- Investigating a resilience-based allocation of multi-WEHs in an uncertain environment to effectively improve the resiliency of the networked MGs;
- Presenting an economic analysis to indicate the effectiveness of multi-WEHs on the networked MGs from the perspective of resiliency against natural catastrophic events.

This paper has been divided into several sections as follows: In the next section, concepts and principles of resilience in the power system are illustrated separately. In Section 3, a problem formulation related to the resilience structure is described. Section 4 is focused on the scenario-based uncertainty method. Section 5 shows the model of the studied method on a sample test MG, as well as discusses and illustrates the results of the provided resilience approach. Finally, the conclusion of this paper is explained in the last section. Figure 1 illustrates the methodology of the work.

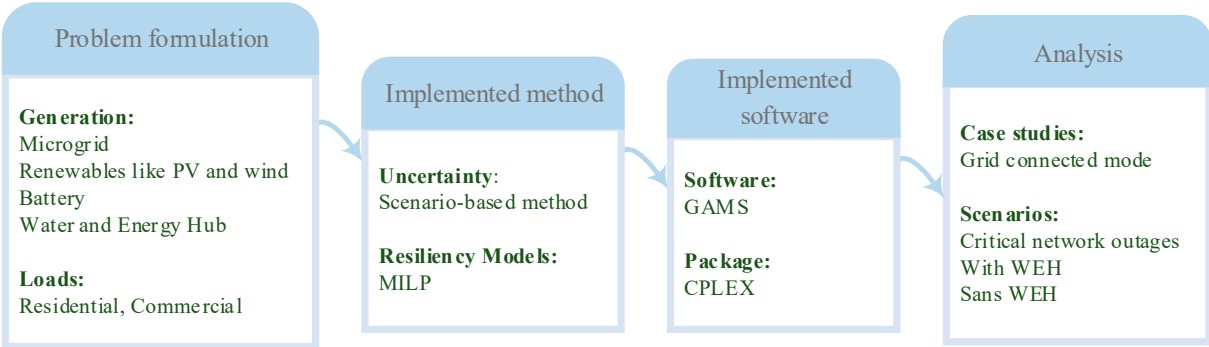

**Figure 1.** Methodology of the work.

## 2. Concepts and Principles of Resilience in Power Systems

Given that resilience is a new and growing concept in the area of power grid analysis and planning and its connection with other power system concepts has yet to be completely resolved, the idea of resilience must be defined. The effectiveness of electrical energy distribution grids in the face of tiny but high-impact catastrophic catastrophes should be thoroughly researched. Engineering equipment is prone to malfunction and damage, and it is unrealistic to anticipate 100 per cent uptime and continuous operation from it. This

section aims to thoroughly study the idea of resilience in power systems in the face of numerous natural catastrophes, as well as to distinguish it from reliability studies.

The related risks and threats must first be defined to explain and assess the system's resilience. In this sense, probable occurrences may be classified into three groups based on the intensity of the effect and the probability (or probability of incidence) [29].

### 2.1. First Category

The first category includes occurrences with a high probability of occurring and whose consequences are restricted and readily quantifiable. Because of their frequency of occurrence, these events have a large database, and their probability of occurrence may be determined by extracting the probability distribution function (PDF) from a large database. The emergency departure rate of a certain production unit or unit of length of a certain kind of transmission line, for example, is known in a power grid, and its values are updated yearly [30,31].

### 2.2. Second Category

The second category of events comprises those that occur seldom and hence lack a large database of occurrences from which to obtain values for their PDF. The repercussions of these occurrences, on the other hand, are roughly predicted. For instance, consider the appearance of a storm in the USA. Although we are unaware of the potential of such a storm, the impacts and implications on infrastructure systems may be approximated since comparable events have occurred in the past. Unidentified or 'Gray Swan' events are the terms used to describe these occurrences. Probability theory does not apply to these occurrences; nevertheless, probability theory, which may be based on expert engineering and theoretical judgments, as well as the mathematical modelling of events, can be thoroughly examined.

### 2.3. Third Category

These are occurrences that have never happened before and whose repercussions are, by definition, unknown. Given the lack of a comparable precedent in these circumstances, discussing the probability of these occurrences is virtually useless; yet, it is conceivable to debate their potential. An earthquake in Tehran, for example, is a scientifically possible occurrence, but the chance of such an occurrence cannot be predicted, and the ramifications of such a calamity are unthinkable. The occurrence of a tsunami in Tehran, on the other hand, is almost inconceivable and should never be mentioned. It is worth noting that following an event in this category, it is moved to the second group since it has happened before and the effects are relatively foreseeable. These are referred to as "unrecognizable occurrences" or "Black Swan" incidents [32,33]. In this paper, events related to Category 2 are examined and evaluated.

The system's performance is shown by the level underneath the resilience diagram (Figure 2). It indicates the amount of load in the network in a vertical-axis distribution system, while the horizontal axis represents time. There are 7 states in the resiliency curve, where various measures could be taken to increase the resilience level, for example, EH preparation in the pre-event status and service restoration applying MGs in the restorative status and post-restoration status.

This study focused on considering an effective resilient scheme to restore the critical loads in a short period after a natural catastrophe when the MG experiences an unpredictable event. By applying the idea of WEHs, there would be a chance of restoring the system by using two sets of WEH systems in the appropriate islanded points to restore the system and critical loads of electricity, heat, and water. This indicates that while the WEH provides some of the energy necessary, it does not fully compensate. Critical loads such as hospitals, gas stations, and military areas are among the loads that are supplied in this period.

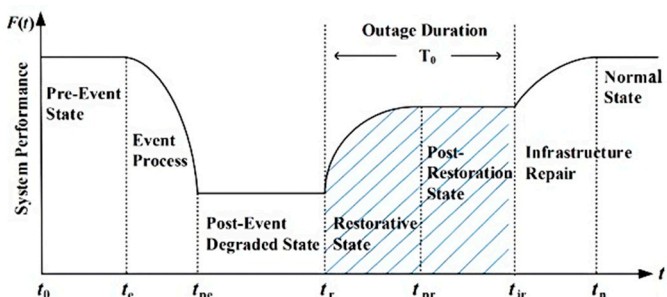

**Figure 2.** Presentation of the resiliency curve.

## 3. Problem Formulation

By integrating the $F(t)$ function over the necessary time period, the distribution system's resilience in the post-accident recovery phase may be calculated. The system's performance is shown by the level underneath the resilience diagram. It indicates the amount of load in the network in a vertical-axis distribution system, while the horizontal axis represents time.

$$R = \int_{t_r}^{t_r+T_0} F(t)dt \tag{1}$$

In Equation (1), $R$ is the criterion for resilience in the period $(t_r, t_r + T^0)$. Increasing and improving the resilience level of the grid is equivalent to increasing the $R$ index. The performance function of the $F(t)$ system is equivalent to the energy supplied to consumers based on their weight priority [34,35].

$$F(t) = \sum_{n \in N^B} W_n P_n^D(t) \tag{2}$$

In Equation (2), $N^B$ is the number of network buses, $W_n$ is the coefficient of load priority, which is identified by the importance of critical loads such as hospitals, airports, etc., and the $P_n^D(t)$ variable is the consumed active power amount in $t$ time. $P_n^D(t)$ is $P_n^D$ amount in the $(t_r, t_r + T^{CL})$ interval in which $P_n^D$ is the active power of the load in $n$ bus, and $T^{CL}$ is the servicing time of loads after the recovery time after a natural event. In fact, $P_n^D(t)$ is zero in the $(t_r + T^{CL}, t_r + T_0)$ interval. $T^{CL}$ will be calculated based on the computational time that EH electrical power, heat, and cooling energy take to reach the critical loads after natural catastrophes. Hence, we can expand the resiliency index as follows:

$$R = \int_{t_r}^{t_r+T_0} \sum_{n \in N^B} W_n P_n^D(t)dt = \sum_{n \in N^B} W_n \int_{t_r}^{t_r+T_0} P_n^D(t)dt = \sum_{n \in N^B} W_n \int_{t_r}^{t_r+T^{CL}} P_n^D(t)dt = \sum_{n \in N^B} W_n P_n^D T_n^{CL} \tag{3}$$

Uncertainties deriving from probable natural catastrophe locations should be incorporated in the model before the accident and while preparing for the EH allocation plan. The quantity of load recovered may vary depending on the circumstance; thus, the system performance function is unpredictable. We must compute the expected value of Equation (3) to analyze and evaluate the success of pre-accident preparation.

$$R_E = E(R) = E[\sum_{n \in N^B} W_n P_n^D T_n^{CL}] = \sum_{n \in N^B} E[W_n P_n^D T_n^{CL}] \tag{4}$$

$R_E$ is the expected value of the resilience index given the uncertainty in Equation (4). The mathematical expectation operator is a performance function that is computed using the probability distribution function and various natural catastrophe situations. As a consequence, the resiliency objective function may be characterized as Equation (5).

$$F(t) = \sum_{n \in N^B} E[W_n P_n^D T_n^{CL}] \tag{5}$$

The higher the expected load recovered after a natural disaster, the greater the network's resilience, according to Equation (5), and, this objective function is equivalent to the ENS in the network, i.e., the less the ENS, the higher the index (5).

In resilience studies, the concepts of all objective functions and indicators introduced are possible based on the load supplied in the minimum time, or the minimization of ENS in the network can be defined as a benchmark. Therefore, several indicators have been defined so far, all of which convey the same concept. The resilience indicators of a distribution MCES are investigated in this part to study the resilience of MCESs. Three energy carriers, electricity, gas, and heat, as well as water, are used to serve electrical and thermal load points, as well as drinking water, in this distribution system. The topology of the structure of the energy distribution system in this MG system, which supplies electrical and thermal loads, is such that if an extreme event occurs in one of the energy carriers, the loads associated with that network may be delivered by another energy carrier. EH equipment is used as an intermediate between energy supply in electrical and thermal networks utilizing a gas network in this system. As a result of the abovementioned explanations, the modified problem's objective function (OF) is provided as follows:

$$OF = min \left[ \underbrace{\sum_t \left( \sum_{elec} ELC_{t,elec} + \sum_{heat} HLC_{t,heat} + \sum_{water} WLC_{t,water} \right)}_{R} + Cost_{Hub} \right] \quad (6)$$

$$Cost_{Hub} = \sum_{t \in \Omega^T} \left( P_t^{CHP} \times price_{CHP} + P_t^{Boi} \times price_{Boi} + P_t^{Bat} \times price_{Bat} + P_{s,t}^{NETM\_H} \times price_{EH} \right. \\ \left. + W_t^{Grid} \times price_{Water} \right) \quad (7)$$

[ELC, HLC, WLC] = [Vector of variables reflecting the Electricity, Heat and Water Load Curtailment]

*Limitations and Constraints*

Equation (8) shows the equilibrium of the electrical power of the introduced EH set. On the left side of this equation, $P_t^E$ is the electrical power demand of the EH, $P_t^{Des}$ is the power consumption of the freshwater desalination unit, and $P_{s,t}^{NETM\_H}$ is the amount of power injected by the hub into the network under study. To the right of this relation, $P_t^C$ is the output power of the CHP unit, $P_t^{bat}$ is the power interchange of the energy storage system (ESS), and $P_t^{boi}$ is the output power of the boiler unit. As noted, the CHP unit is responsible for supplying electricity and heat at the same time, and $\eta_{chp}^{GtoE}$ is the coefficient of electricity supply, which, in this paper, is regarded as 45% of its total production capacity. As noted, the boiler is responsible for supplying heat, during which the gas is consumed as an input and the generated heat as its output product. The amount of consumption and production of this unit is regarded in the balance constraints. For example, if there is a need for a high power supply per hour from the CHP unit and the battery unit, then the task of supplying heat load power is the responsibility of the boiler unit, which increases the consumption and production using balance constraints. The $\eta_{boi}^{GtoE}$ is the output power of the boiler unit. $\eta_e^T$ is the efficiency of the transformer.

$$P_t^E + P_t^{Des} + \eta_e^T \sum_{s \in \Omega^s} P_{s,t}^{NETM\_H} = \eta_{chp}^{GtoE} P_t^C + P_t^{bat} + \eta_{boi}^{Gtoe} P_t^{boi} \quad , \forall t \in \Omega^T \quad (8)$$

Equation (9) indicates that the power exchanged between the EH and the MG is within its allowable range. Equations (10)–(12) are due to the charge and discharge limit of the ESS of EH at any point in time.

$$\underline{P}_t^{EH} \leq \sum_s P_{s,t}^{NETM_H} \leq \overline{P}_t^{EH} \quad , \forall t \in \Omega^T \quad (9)$$

$$\underline{Eb}^{bat} \leq Eb_t^{bat} \leq \overline{Eb}^{bat} \quad , \forall t \in \Omega^T \tag{10}$$

$$Eb_t^{bat} = \left(1 - ES_e^{loss}\right) Eb_{t-1}^{bat} + P_t^{bat} \tag{11}$$

$$\frac{1}{\eta_e}\underline{P}^{bat} \leq P_t^{bat} \leq \frac{1}{\eta_e}P^{bat} \quad , \forall t \in \Omega^T \tag{12}$$

$$P_t^{bat} = P_t^{ch} - P_t^{dch} \tag{13}$$

In Equation (10), $Eb_t^{bat}$ is the capacity of ESS (kWh), and $\overline{Eb}^{bat}$ and $\underline{Eb}^{bat}$ indicate the maximum and minimum storage capacity of energy, respectively.

Based on Equation (11), the energy stored in moment t is equal to the energy stored in the prior hour plus the amount of energy lost and the amount of charge and discharge of the ESS in time t. In Equation (11), $ES_e^{loss}$ is the loss coefficient of electrical ESS. Equation (12) shows the amplitude limit of the amount of energy stored and discharged in the battery unit of the EH. This equation illustrates the efficiency of charging and discharging the ESS.

Moreover, in Equation (13), the exchanged power of electrical ESS in each moment is $P_t^{bat}$ as the amount of charge power minus $P_t^{ch}$, the amount of battery discharge power $P_t^{dch}$.

$$P_t^{Heat} = \eta_{chp}^{GtoH}P_t^C + \eta_{boi}^{GtoH}P_t^{boi} \quad , \forall t \in \Omega^T \tag{14}$$

Equation (14) indicates the relation of thermal energy balance in each moment and guarantees the EH thermal demand in each moment. In this relation, $P_t^{Heat}$ is the demand amount of thermal energy at $t$ moment, $\eta_{chp}^{GtoH}$ is the exchange return of natural gas to thermal energy in the CHP unit, and $\eta_{boi}^{GtoH}$ is the exchange return of natural gas to thermal energy in the boiler unit.

$$P_t^{Gas} = P_t^C + P_t^{boi} \quad , \forall t \in \Omega^T \tag{15}$$

Equation (15) expresses the whole gas consumption of the network, which is the sum of the input gas of the CHP units and the boiler unit.

$$\eta_e^T \left(\sum_s P_{s,t}^{NETM\_H}\right) \leq Cap^{Tr} \quad , \forall t \in \Omega^T \tag{16}$$

$$\eta_{chp}^{GtoH}P_t^C \leq Cap^{CHP} \quad , \forall t \in \Omega^T \tag{17}$$

$$\eta_{boi}^{GtoH}P_t^{boi} \leq Cap^{boi} \quad , \forall t \in \Omega^T \tag{18}$$

The output power of the EH unit in Equation (16) must be less than the transformer capacity. In this relation, $\eta_e^T$ is the transformer electrical power return, and $Cap^{Tr}$ indicates transformer capacity (kW).

Moreover, Equations (17) and (18) indicate the amount of output-permitted power of the CHP unit and boiler unit. In this equation, $Cap^{CHP}$ is unit capacity (kW), and $Cap^{Boi}$ shows transformer capacity (kW).

$$V_t^{ST} = V_{t-1}^{ST} + W_t^{OD} + W_t^G - W_t^{Out} \quad , \forall t \in \Omega^T \tag{19}$$

$$V_t^{DT} = V_{t-1}^{DT} + W_t^{ID} + W_t^{OD} \quad , \forall t \in \Omega^T \tag{20}$$

Equations (19) and (20) indicate the desalination unit water balance constraints. In this equation, $V_t^{ST}$ is the reservoir of drinkable water in $t$ time $(m^3)$, and $V_t^{DT}$ is the reservoir volume of desalination unit in $t$ time $(m^3)$. Moreover, $W_t^{Out}$ is the demand amount of system water load in which the $W_t^{Out}$ amount includes the reservoir output of the desalination unit $\left(W_t^{OD}\right)$ and the amount of poured water into the network $W_t^G$ in which

the water capacity of the desalination unit is limited by Equation (20). In Equation (20), $W_t^{ID}$ is the input water amount to the reservoir of the desalination unit in t time.

$$\underline{W}^{ID} \leq W_t^{ID} \leq W^{ID} \quad , \forall t \in \Omega^T \tag{21}$$

$$P_t^{Des} = W_t^{ID} \cdot CF^{Des} \quad , \forall t \in \Omega^T \tag{22}$$

The amount of water entering the desalination unit is limited by Equation (21). In fact, the amount of seawater inlet volume that enters the desalination unit $\left(W_t^{ID}\right)$ is limited by Equation (21). In this equation, $W^{ID}$ and $\underline{W}^{ID}$ are the maximum and minimum water input capacities of the desalination unit, respectively.

Furthermore, the desalination unit consumes a power that linearly is related to the amount of water entering the desalination unit, which is indicated in relation to Equation (22). In this equation, $CF^{Des}$ is the energy consumption coefficient per unit of desalination water (kW/L). Moreover, in all relations related to the water sector, the suggested EH system is the unit of water volume in terms of cubic meters $\left(m^3\right)$.

## 4. Scenario-Based Uncertainty Model

The scenario-based approach is known as a tool for modeling uncertainty associated with the parameters of a problem simply and effectively. The input parameters of the optimization problem include electrical, heating, and water demands as well as electricity, natural gas, and water prices. The proposed uncertain parameters are modelled using the Gaussian probability distribution function (PDF). For modeling, the standard deviation of the uncertain parameters is 0.15 of the mean of parameter values. Figure 3 represents the flow chart of the scenario-based method. By simulating the Roulette Wheel approach (commonly known as the fitness proportion selection as a way to randomly select from a given list of weighted inputs), 1000 scenarios are obtained. The data value for each parameter is uncountable and undefined. Based on the high number of scenarios and them being time consuming to process, the generated scenarios were reduced to the 10 best possible ones based on the backward reduction algorithm [22]. It should be noted that in reducing scenarios, the closest scenarios to the main scenario were selected, while others that were far away from the main scenario were eliminated. The uncertainty existing in the energy consumption of demand loads on a typical day of the year is depicted in Figure 4. It shows the input-generated scenarios of electrical load, heat load, and water load and the real-time electricity price on a typical day of the year.

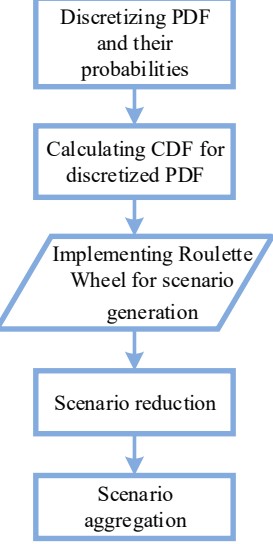

**Figure 3.** Flow chart of Scenario-Based approach [22].

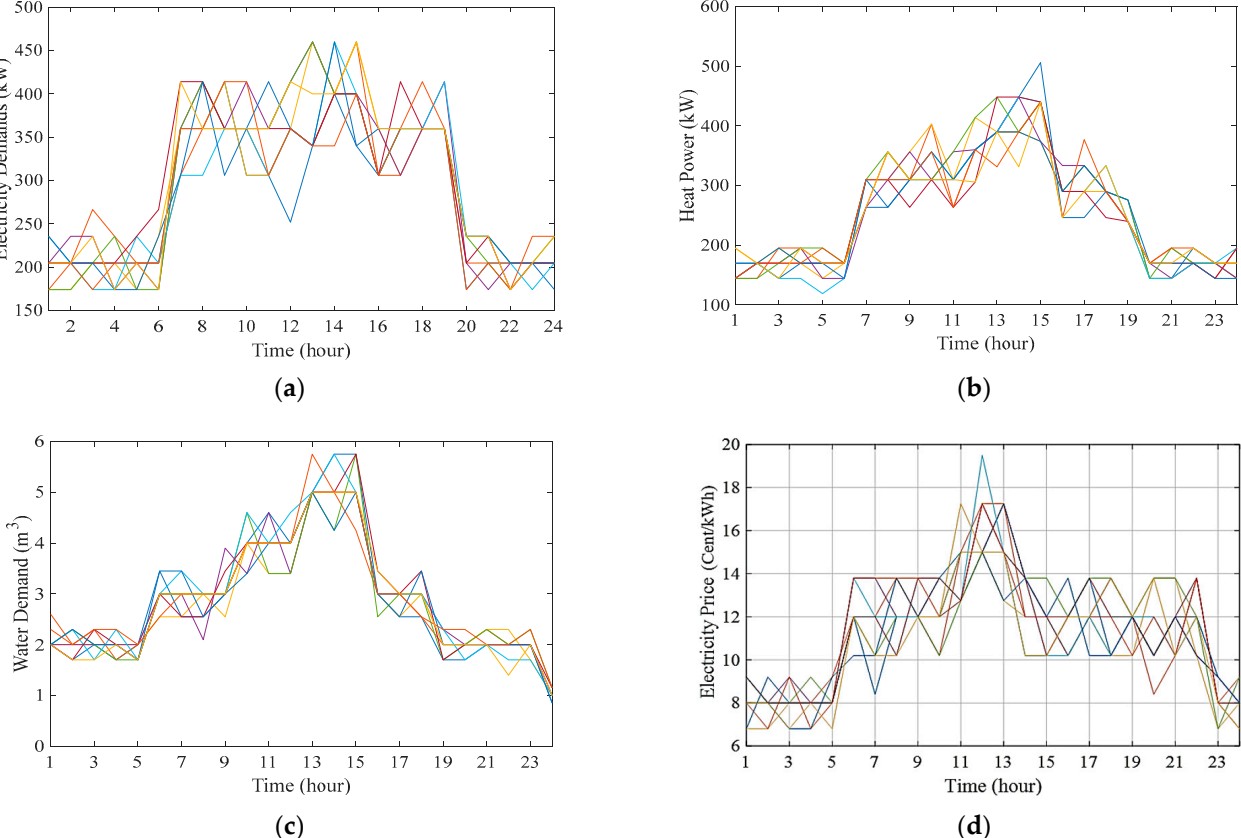

**Figure 4.** Presentation of the input generated Scenarios on a typical day of the year: (**a**) Electrical load scenario; (**b**) Heat load scenario; (**c**) Water load scenario; (**d**) Real-time electricity price scenarios. Each plot shows a generated scenario.

## 5. Simulation Results and Discussion

After a natural disaster, such as a storm, the goal of resilience research is to respond to critical loads in the quickest time feasible. Critical loads recognized in the network may be responded to in the shortest possible time based on the condition of the network after the damaging occurrence, as mentioned in the previous section, by considering the EH systems and optimum design of the MCES. As a result, determining the dimensions and the best placement of this type of MCES in MG is a fundamental problem to increase network flexibility. As a result, one of the alternatives for boosting the network's durability and resilience is to deploy MCESs known as EHs, which may be planned to handle the greatest number of essential loads. The outcomes of the existence of two sets of combined energy production (water and EH) to strengthen the resilience of the sample network against natural catastrophes are explored in this chapter based on the descriptions and connections presented in previous sections. Moreover, the suitable position of these two sets was replaced by conceivable catastrophe scenarios, with the ultimate objective of improving MG resilience and lowering the MGs' blackout rate.

Figure 5 depicts a single-line schematic of the 24-bus networked MG to explore the influence of the existence of the EH on the enhancement of resilience, as previously stated. In this part, two sets of combined EH sites are optimized based on the catastrophe scenario, and the effect of their existence after a natural catastrophe is completely detailed. In reality, after a natural catastrophe and the loss of grid lines, the energy exchange between the EH and the grid, as well as the creation of an energy balance, results in the grid supplying electrical energy to key loads (such as hospitals, airports, and so on). This improves the MGs' resilience and endurance.

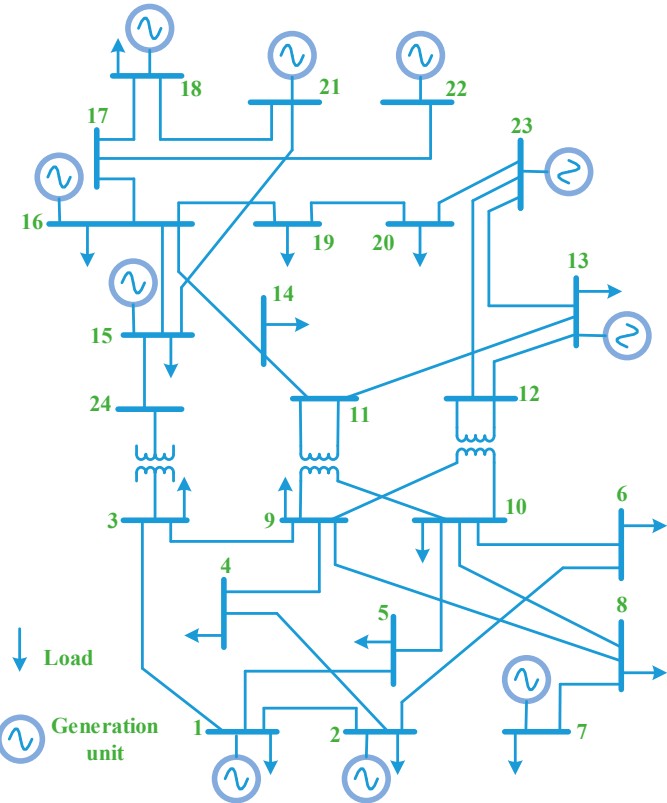

**Figure 5.** The diagram of the 24-bus trial system.

As previously stated, to strengthen resilience, we aim to locate two sets of smart WEHs during a probable natural catastrophe scenario. Reference [36] provides information about the MG and EH set. A CHP unit, a boiler, a battery unit, and a water desalination unit are included in each of these sets. Figures 6 and 7 depict the load profiles of these two sets of EHs over of 24 h after the natural catastrophe, which include the thermal load, water consumption, and electric charge of these two sets of EHs. In this study, it was assumed that the heat and water system was destroyed after the natural catastrophe and that the heat and water consumption of the MG buses is supplied only through WEHs.

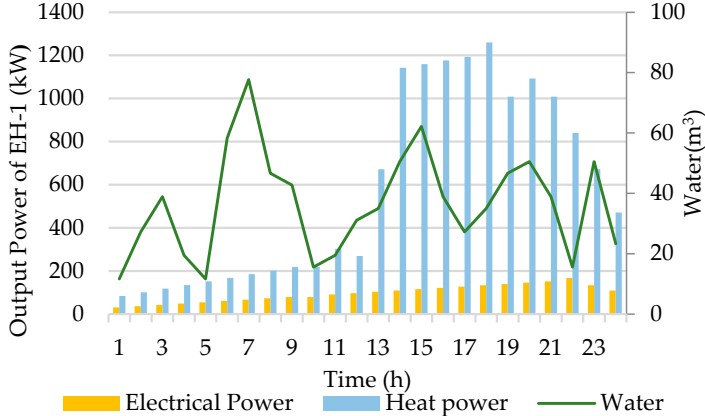

**Figure 6.** Load profile of EH-1.

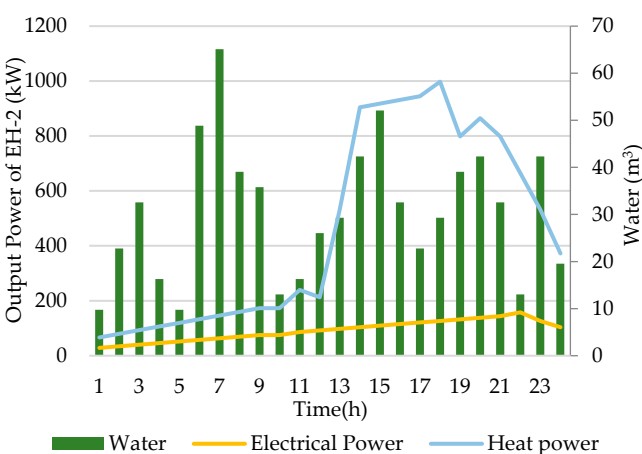

**Figure 7.** Load profile of EH-2.

This section describes the simulations and their outcomes. GAMS and MATLAB software were used for the simulation. The mixed integer linear programming (MILP) technique is carried out in GAMS 23.6 and solved through the CPLEX solver on a PC with an Intel Core Due 2.2 GHz processor and 2 GB of RAM in less than 7 s. In this method, the output of the GAMS software is fed into the program in the form of data at regular intervals, and the appropriate diagrams are generated when all of the information has been entered into the software. It should be noted that the effect of the performance of these two sets of EHs on reducing network blackout due to the occurrence of a probable accident is investigated using different scenarios of its occurrence. In all situations, four possibilities are explored to meet the problem's objectives, with the reasons for selecting all four scenarios detailed below. The following are four conceivable natural catastrophe scenarios:

1. First scenario: the destruction of generator no. 10 as a unit of energy resources in the MG due to the occurrence of a natural catastrophe with this generator being connected to bus no. 7 according to Figure 5;
2. Second scenario: the destruction and disconnection of line 12 related to a natural disaster, which connects bus number 8 to bus number 9 according to Figure 5;
3. Third scenario: the effect of natural disasters and the destruction and disconnection of line 22, which connects bus 13 to bus 23;
4. Fourth scenario: the effect of the simultaneous occurrence of two faults on lines 21 and 22 related to the occurrence of a terrible disaster in the sample MG. Line 21 connects bus 12 to bus 23.

For this purpose, four possible natural catastrophes were considered in this section to reduce blackouts caused by the natural catastrophes. The effect of the presence of two proposed sets of EHs on the rate of clear blackouts as well as the economic impact of their presence during natural catastrophes is determined. The scenarios are based on a real event to show the level of network failure. In each scenario, the destruction of each network element is considered based on the severity of that event. We tried to simulate all cases of failure of network equipment after an event so that the problem is closer to reality.

*5.1. Analysis of the Effect of WEH on MG Resilience*

We analyze the impact of two sets of EHs on lowering the rate of blackout and enhancing MG resilience due to natural catastrophes in this section, taking load uncertainty into account. To accomplish this, we must first calculate the position of these two sets of EHs in the grid with the abovementioned potential scenario to minimize the amount of blackout while still being able to interchange electrical power with the grid after the natural catastrophe. The ideal supply of all energy in this EH is seen together to lower the rate of blackouts. This is in addition to discovering acceptable areas for energy exchange between these two sets of EHs with the grid. In other words, during the occurrence of a destructive

natural catastrophe, the optimal exchange rate of electrical energy between each set of EHs and the grid is calculated in such a way that the constraints on the balance of heat load supply, as well as power supply for freshwater production units, are met. Figure 8 shows the WEHs allocation results.

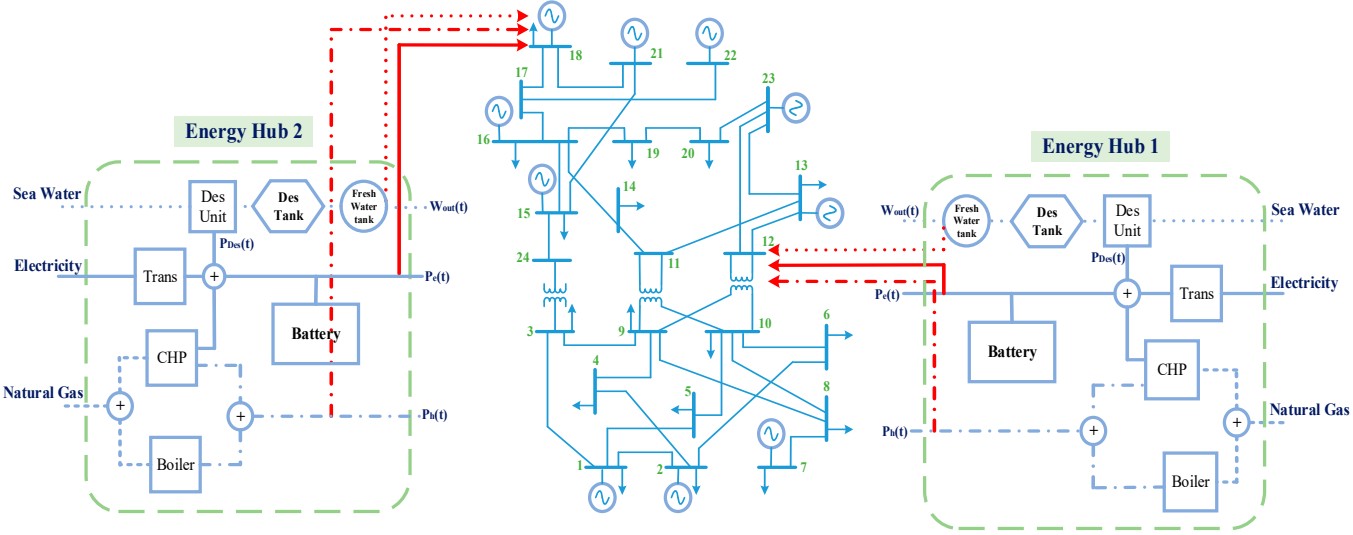

**Figure 8.** Illustration of the WEH resilience allocation results in the MG.

The best sites for these two sets of EHs, according to the findings of the process of optimal placement of WEHs to strengthen the network's resilience against natural catastrophes, are buses 12 and 18. The influence of the existence of these two sets on decreasing blackouts in this situation is evaluated analytically in this section, as well as the outcomes of lowering the blackout due to the occurrence of an assumed natural catastrophe with and without these two sets in four scenarios. CHP units, batteries, boilers, and water desalination units are all included in both planned energy centers. A co-generation unit of electricity and heat (CHP), a boiler, a battery unit, and a desalination unit are among the production units examined in these two complexes (EH). The first unit of consumption is the unit of freshwater production, which, while being a production unit, is a consumer whose quantity of energy consumption must be taken into account in the balance. The CHP and battery unit are in charge of this unit's power supply. The battery unit is another entity that is both a producer and a consumer because this unit must be charged in certain situations, which is the duty of the CHP. It should be emphasized that in this study, the charging times of this unit were determined ideally. Finally, the load of the electrical network, which is provided via the interchange of electrical power between the network and the set of EHs, is the third unit that is defined for these two sets of EHs as consuming electrical power.

After defining the production and consumption units, the impact of having two combined energy production units, which is the paper's major goal, on enhancing network resilience was represented analytically, which was supported by simulation results. The quantity of electrical energy exchange between these two sets of EHs and the network is represented separately in Figure 9 for this purpose. According to Figure 9, these two sets of EHs provide part of the network power demand at diverse times during the 24 h. Moreover, the power supply is provided at various times of the day and night so that, in the event of a network accident or incident, a portion of the network demand will be met; or, at least, according to the single-line view of the grid (Figure 8), in the event of a network accident, the loads around these two sets of EHs in bus number 18 itself or buses 13, 10, and 6, which are close to bus 12 and are related, will be provided. As a consequence, the index discussed in the previous section will improve, and the system's resistance to natural catastrophes will improve. Additionally, the outcomes of the production and consumption

of various units from these two sets were retrieved independently. Figure 10 shows the input power of the CHP unit and the boiler of the two sets throughout 24 hours.

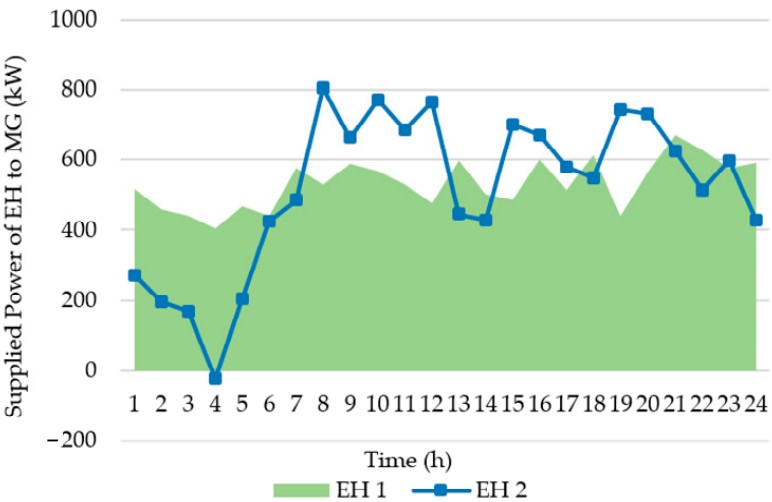

**Figure 9.** The amount of power exchanged between the EH sets and the MG during 24 h.

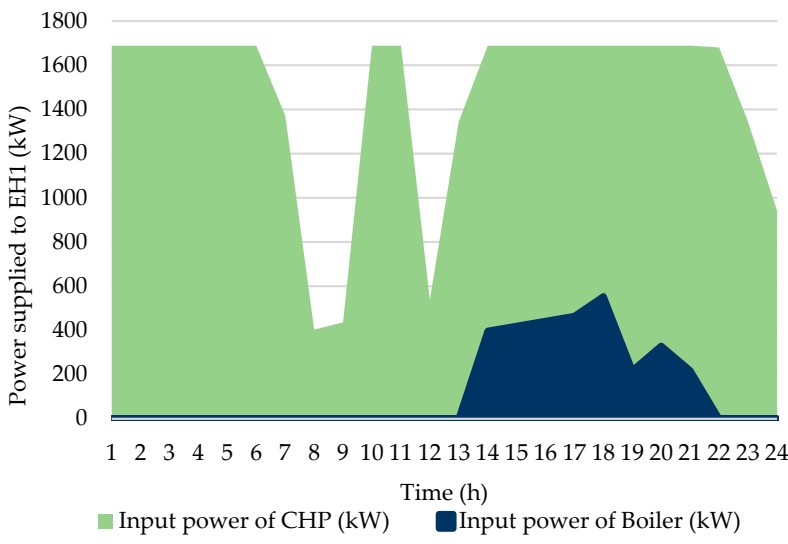

**Figure 10.** The amount of input power to EH-1 during 24 h.

In Figures 11 and 12, in times when the battery power is negative, it means that the battery is being charged. In times when the battery power is positive, the power is transmitted to the consumer. Charging and discharging times are optimal and are based on equilibrium constraints. Moreover, the amount of water production of the water desalination unit during 24 h for both sets of EH is shown separately in Figure 13.

The minimal quantity of electrical energy exchange between the grid and the EH validates the grid's power consumption, as seen in the figures. In terms of electric power, a water desalination unit is a unit that consumes electricity; hence, in Figure 14, the power consumption of the two sets of water purification units throughout 24 h a day, as well as the quantity of gas used per unit, is shown. Figure 15 shows the power consumption units for boilers and CHP.

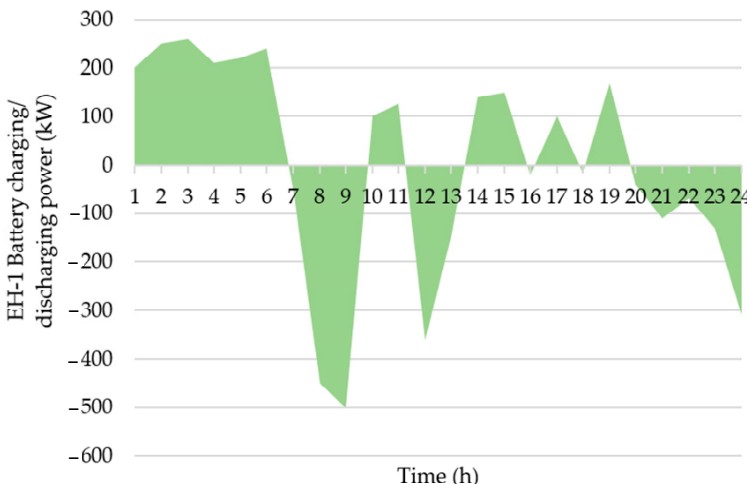

**Figure 11.** Charging and discharging power of the Battery of EH-1.

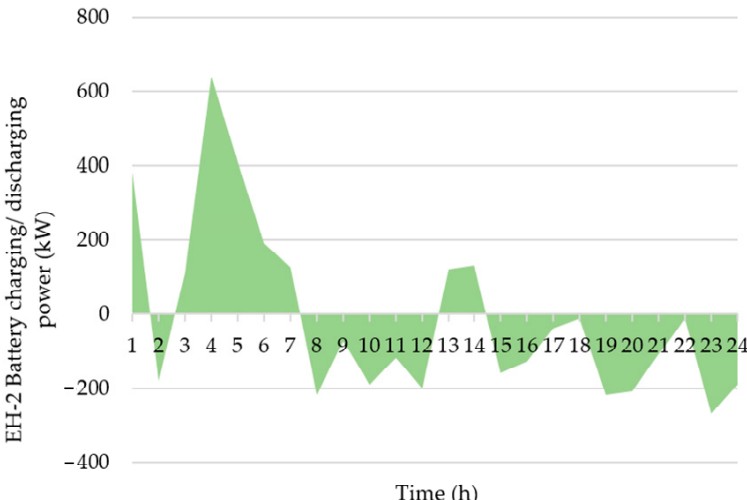

**Figure 12.** Charging/Discharging power of the Battery of EH-2.

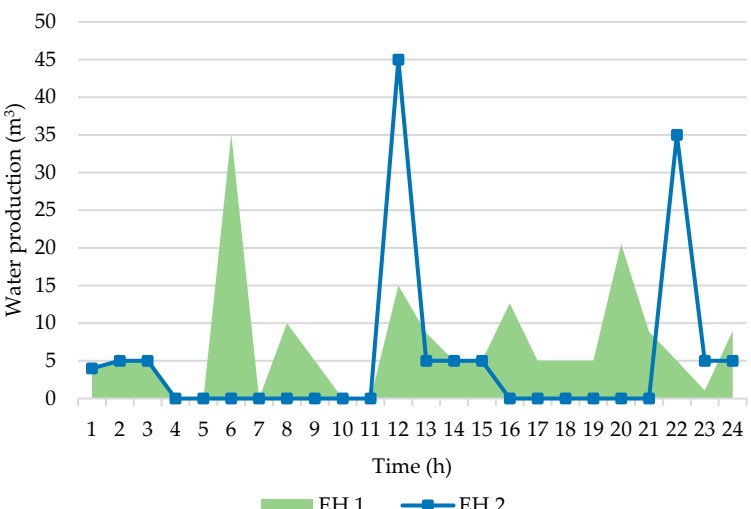

**Figure 13.** Water production of the water desalination unit during 24 h.

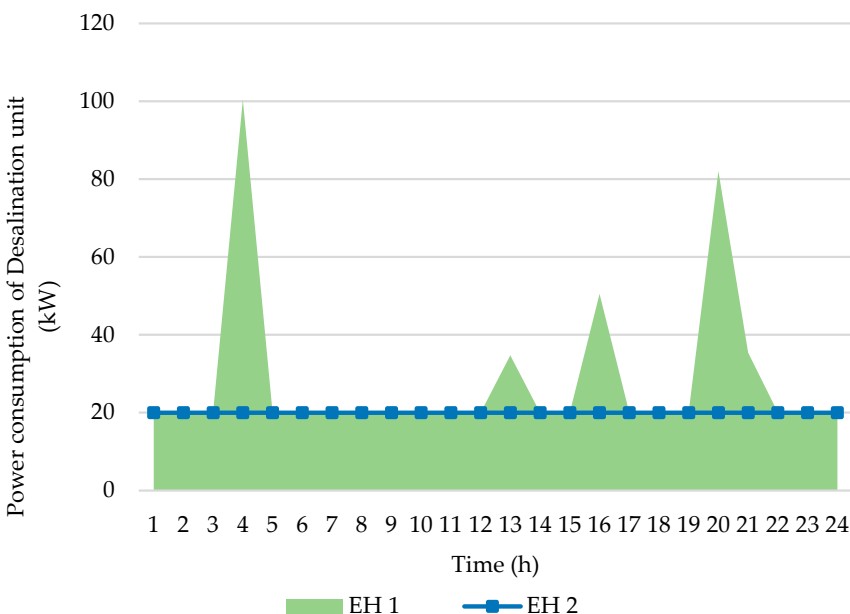

**Figure 14.** Power consumption of the water desalination unit during 24 h.

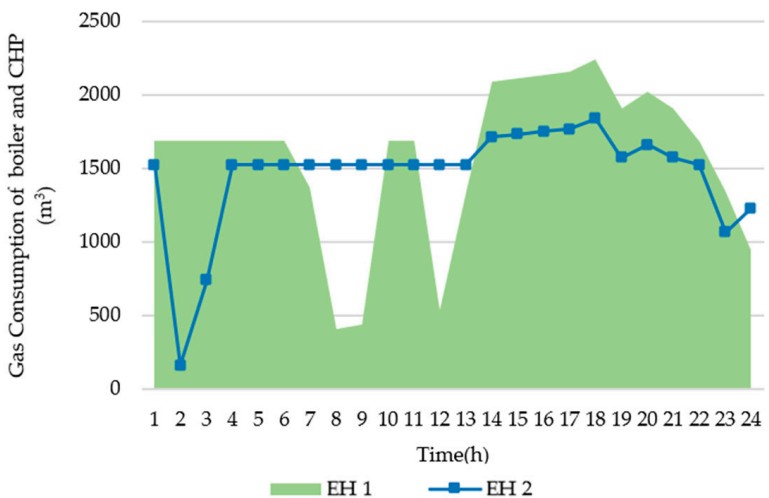

**Figure 15.** Gas consumption of the CHP and boiler unit of EH-1 and EH-2 during 24 h.

According to the results obtained after the natural catastrophe in the MG, it can be demonstrated that in this paper, observing all constraints, without interruption in supplying other network loads such as thermal loads, and simultaneously supplying the demand of water consumers and electrical power, both units of the EH in energy exchange with the MG studied, which involved and consequently increasing the level of resilience and reducing the number of network nodes, are capable of increasing the network's resilience and reducing the number of network nodes. In the MG, the amount of load lost due to the storm and natural catastrophe plays an active part.

We analyzed the impact of the existence of these two sets of EHs on enhancing network resilience as an example to better understand the problem. According to the obtained data, if we consider at 10 h and also want to assess the efficiency of the EH set 2 in this time, the amount of electric charge demand in the hub set energy is equivalent to 74.88 kWh, as shown in Figure 7. Meanwhile, according to Figure 14, the consumed power of water desalination unit is 20 kW. According to Figure 9, the quantity of electrical power that this set exchanges with the network at the same time is 591.58 kW. The EH set consumes a total of 2 times 686.46 kWh of electricity. A considerable quantity of power is exchanged with the network from this amount of electricity. As a consequence, the same amount of

electrical power must be produced by this unit for it to appropriately perform its active part at that time. Of course, this does not imply that it must be connected to the electrical grid, but it does imply that the balancing rules must be followed to the letter. Following a natural catastrophe such as a storm, the same balance and energy exchange may strengthen the distribution system's resilience. Meanwhile, as can be seen in Figure 10, the total output power of CHP unit of the EH2 is 1520 kW. The total thermal and electrical power equals this quantity of energy. As a consequence, given that the coefficient of supply of electric power of the CHP unit in this paper is 45 percent, the total generated electric power of the CHP unit at this hour is 684 kW. The battery unit, which is drained at 10 o'clock according to Figure 10 and has a producing power of 2.46 kW, is also a unit that may play a part in generating power in this clock. As a result, the total electrical power created by EH-2 over 10 h is 686.46 kWh, which is the same as the total electrical power used by EH-2 during the same period.

### 5.2. Economic Validation for Applying Two Sets of WEHs in the MG to Improve Resilience

One of the main reasons for employing EH sets, as explained in the previous sections, is that they may be utilized in an urban capacity in the event of a natural or unanticipated disaster in the operating network.

Given that one of the most significant issues in the debate on energy supply in today's society is the price and cost utilized in each study, the economic reason for employing the suggested set of EHs to increase MG resilience is examined in this part. We demonstrate that, in addition to technical reasons, this capability may be economically feasible in crucial instances such as natural catastrophes. The presence of WEHs, according to the results obtained in the previous part, is effective in reducing energy supply and improving resilience caused by natural catastrophes because part of the network power is provided by these sets, which is effective at least in local loading (if properly located) in the event of a natural disaster such as a storm. After a natural catastrophe, the same problem of local power supply decreases the number of unsupplied loads in the network and minimizes blackouts, boosting the system's resilience to natural catastrophes. For this purpose, four possible extreme event scenarios were considered in this section to locate the WEHs to reduce blackouts caused by the natural catastrophe, in the context of which the effect of the presence of two proposed sets of EHs on the rate of clear blackouts, as well as the economic impact of their presence, was determined. The purpose of selecting the abovementioned four scenarios is to take into account all facets of the resiliency study. In reality, in these four scenarios, an effort was made to study both the effect of leaving a line due to a natural catastrophe at two separate places in the MG and the effect of leaving a unit producing power, as well as the impact of natural catastrophes. The network develops such that the probability of two network problems when an event strikes at the same time is considered. The results of the existence of these two sets of EHs in lowering the quantity of unsupplied energy and enhancing network resilience by executing each scenario are shown in Table 2, which also compares and shows the effective influence of EHs on improvement. In all four scenarios, there always was a deficiency. Moreover, the deficiency is managed such that critical infrastructure such as hospitals are prioritized.

**Table 2.** The comparison of ENS between two states.

| Scenario | ENS (kWh) | |
|---|---|---|
| | **With Considering WEH** | **Without Considering WEH** |
| First | 41.2 | 102.3 |
| Second | 83.9 | 189.5 |
| Third | 95.5 | 214.5 |
| Fourth | 114.5 | 248.6 |

The lack of an EH in the grid is contrasted with system resilience scenarios. This represents a considerable drop in the number of blackouts over the year, or even over time. The investment cost for the construction and operation of the EH established to make the MG robust is acceptable when we explain the reduction in blackouts into a unit of cost. Table 3 illustrates the comparison of the resilience function in the various cases in this paper.

**Table 3.** The comparison of resilience function in the different cases.

|  | ENS (kWh) | Investment Cost (USD) | Benefit of Selling Power to the grid (USD) | Objective Function (USD) |
|---|---|---|---|---|
| Considering WEH | 41.2 | 18,286.4 | 11,584.4 | 141,333,546 |
| Without considering WEH | 102.3 | - | - | 168,523,563.5 |

The uncertainty of MG and EH loads, including thermal and electrical loads, as well as water usage, is discussed in this part. Naturally, given the uncertainty, the investment cost of implementing the complicated EH implementation plan would rise, since the quantity of output must be forecasted for the worst-case scenario under these circumstances. On the other side, this forces the EH complex to play a bigger role in power supply following a natural catastrophe, which increases the economic cycle associated with energy sales. The findings of the optimal allocation of WEHs, taking into consideration the uncertainty induced by electrical and thermal demands, as well as water usage, are depicted in Table 4.

**Table 4.** The comparison of resilience function in the different situations of uncertainty.

|  | With Considering Uncertainty | Without Considering Uncertainty |
|---|---|---|
| ENS (kW) | 33.8 | 41.2 |
| Investment Cost (USD) | $21,151$ | 18,286.4 |
| Benefit of Selling power to grid (USD) | 12,992.1 | 11,584.4 |
| Objective Function (USD) | $1.4154 \times 10^8$ | $1.4133 \times 10^8$ |

As the outcomes presented in Table 4 show, while the cost of investment has increased in this scenario (first scenario), instead of increasing the effect of the WEH set on power supply in critical and definite conditions after a natural catastrophe, the profit from power sales to the MG has also increased. This makes the investment cost less obvious over time, balances the entire cost, and makes the investment economically feasible.

As a consequence, while preparing for the execution of such plans, it is preferable to incorporate the load's uncertainty so that the problem's circumstances are more realistic. Furthermore, given the load's uncertainty, if the amount of load changes can be predicted and the load can be supplied locally with the most optimum mode, more control may be gained in this area. Because increased investment in this subject also covers the worst-case scenario, this is the case. As a result, because we are not constantly in the worst condition, the operator will be more willing to limit the number of blackouts after a storm mistake. This benefit should also be included in Table 4, as well as its influence on the overall cost.

As a consequence, following a natural catastrophe, the findings of the existence of EH sets in the MG were studied. Given that the initial purpose of this paper is to increase the MG system's resilience to natural catastrophes, the outcomes of implementing the recommended EH presence plan reveal that using this capacity may help to improve the system's resilience. The effectiveness of a generative network is that it reduces the quantity of unsecured energy in the network due to severe disasters. It also has economic reasons. Furthermore, the usage of this capacity may have a variety of advantages for the network; nevertheless, among the numerous benefits that the execution of this paper can have, the influence of utilizing this capacity on one of the key indicators is discussed in this paper. The rate of blackouts and the decrease in unmet supply were used to assess power quality.

Reducing this index improves network resilience. The cost of investment in this area is justified, and it also promotes customer satisfaction in the field of sustainability, according to the results of the profit from the effective and optimum usage of these sets in the network resilience process. Power supply under all circumstances, including severe conditions such as those induced by natural catastrophes, is ensured.

## 6. Conclusions

This paper suggested an enhanced resilient scheme according to WEHs within an MG. This study focused on considering an effective resilient scheme to restore the critical loads in a short period after a natural catastrophe when the MG experiences an unpredictable event. By applying the idea of WEHs, there would be a chance of restoring the system by using two sets of WEH systems in the appropriate islanded points to restore the system and critical loads of electricity, heat, and water. In this regard, an effective CPLEX solver of GAMS software was used to find the optimal allocation of EHs considering the cooperation of multi-energy infrastructure in the MG. For this purpose, different scenarios were considered for assessing the resiliency of the network against a natural catastrophic event that causes serious damage to the network by analyzing the ENS factor. Since the MGs are incorporated with RESs, the scenario-based method is presented to capture the uncertainty associated with the electrical, heat, and water demands and electricity, natural gas, and water prices. It was found that resilience will be increased if MCE is integrated into an effective and resilient framework such as WEH considering the various statuses of the system. In this work, we assumed that the heat and water system outage is caused by the natural catastrophe and that the heat and water consumption of the MG buses is supplied only by the WEHs. Moreover, the allocated WEHs can adequately supply the electrical, water, and thermal demand loads after the natural catastrophe throughout the day. Moreover, it was shown that the suggested scenario-based approach as an uncertainty scheme could consider the uncertainty of electrical, heat, and water demands of WEHs in the MG. In this study, it was assumed that the heat and water system was destroyed after the natural catastrophe and that the heat and water consumption of the MG buses is supplied only through WEHs. Further research should be performed on heat and water networks during natural catastrophes because considering all networks at the same time helps to improve network resilience faster.

**Author Contributions:** Conceptualization, Data curation, Formal analysis, Software, Investigation, Resources, and Writing—Original Draft carried out by S.S.; Project administration, Supervision, Validation, Writing—Review and Editing, Visualization, Methodology, and Funding Acquisition performed by M.D. and P.S. All authors contributed equally to this paper. This research paper is contributed to by the authors mentioned above. All authors have read and agreed to the published version of the manuscript.

**Funding:** This research received no external funding.

**Institutional Review Board Statement:** Not applicable.

**Informed Consent Statement:** Not applicable.

**Data Availability Statement:** Data are contained within the article.

**Acknowledgments:** We would like to thank Mahmoud Roustaei from Shiraz University of Technology for his useful guidance in the various phases of preparing this research to improve the overall scientific quality of the paper.

**Conflicts of Interest:** The authors declare no conflict of interest.

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
