# Peer review of "Microgrids Resiliency Enhancement against Natural Catastrophes Based Multiple Cooperation of Water and Energy Hubs"

_smartcities, doi:10.3390/smartcities6040082_

Round 1
Reviewer 1 Report
Authors propose a method to enhance resiliency in integrated energy systems considering two set of water and energy hubs. The paper is interesting however there are some concerns as follows:
1- Please revise the Abstract, and present some of the achievements of this study.
2- English-wise, the paper should be checked.
3- The literature review should be updated with some recent works which model integrated energy systems and associated risks.
https://doi.org/10.1016/j.apenergy.2022.119315
https://doi.org/10.1016/j.est.2023.107103
https://doi.org/10.3390/en11102573
4- Please pay attention to SI units and it is recommended to provide it where a variable is introduced. Besides, kw should be revised to kW.
5- The CHP model seems inappropriate. Please compare it with the model introduced in https://doi.org/10.1016/j.energy.2023.127084, and explain why you have used this model.
6- The uncertainty modeling is unclear. Please provide a reference or explain your method.
7- Please explain more about the methodology used to solve the proposed problem and if applicable provide a flowchart. What software or solvers are employed?
8- How is the convergence and time needed for the proposed method? What are the merits of this method compared to other methods?
9- The prospects for further research are not given.
Author Response
Dear Reviewer
Authors sincerely appreciate the careful attention to the present work. Comments that we received are really precise and explicit and are addressing the thorough insight of reviewers to the intended subject. Suggestions and mentions of reviewers really helped us to improve the manuscript and to provide a better paper of higher quality. To be clear, all changes and modifications are colored in the new version of manuscript. Extensive adjustments have been utilized to achieve the requirements which were mentioned by reviewers. All comments are responded in the attached file.
Please see the attachment file.

Reviewer 2 Report
Please find the review attached.

Author Response

(The authors gave the same response as above.)

Author Response

(The authors gave the same response as above.)

Round 2
Reviewer 1 Report
No further comments.
Author Response
Reviewer # 1:
No further comments.
Authors’ Response: Firstly, we thank you for your time to review our paper and also valuable recommendations in the first round to improve the paper. We are glad to hear your positive feedback.
Reviewer 2 Report
Please find the review attached.

Author Response
Authors’ Response to Reviewers’ Comments
Reviewer # 2:
I thank the authors for addressing some of the comments from the previous review. While the readability of the paper has greatly improved, the primary concerns have not been addressed:
Authors’ Response: Firstly, we thank you for your accurate attention and valuable recommendations. We are glad to hear your positive feedback and would like to thank you for giving us constructive suggestions to improve the quality of the paper.
Comment # 1: How does this approach compare to those mentioned in the literature? It is necessary to show quantitatively that for the same 24-bus trial system that this method provides more resiliency that other methods. Why were two EHs chosen? Does two only work for the 24-bus system?
Authors’ Response: Thank you very much for your comment. After carefully reviewing the related literature, we came up with the idea of multi water and energy (WEH) for resiliency improvement of the networked MGs. For this purpose, different scenarios were considered for assessing the resiliency of the network against natural catastrophic event that causes serious damages on the network by analysing the energy-not-supplied (ENS) factor. Since the MGs are incorporated with renewable energy sources (RESs), scenario-based method is presented to capture the uncertainty associated with the electrical, heat, water demands and electricity, natural gas and water prices. In this work, we assumed that the heat and water system is out caused by the natural catastrophe, and the heat and water consumption of the MG buses is supplied only by the WEHs. This concept has not been followed by related literature. Also, it should be mentioned that the studied 24-bus networked MGs case is chosen to perform the simulations on it and demonstrate he effectiveness of the work. This approach is applicable on other networked MGs that have the RESs and are able to host WEHs.
Comment # 2: It is also necessary to show how is the method generalizable to systems other than the 24-bus one? A methodology that the reader can follow and apply it to other system would show the contribution of the authors.
Authors’ Response: Thank you very much for your comment. A methodology is provided and added to the new version of the paper.
Comment # 3: Line 125: “Another weakness of these references is that it does not consider other equipment in the power grid.” Could the authors elaborate on the ‘other equipment’?
Authors’ Response: Thank you very much for your comment. The sentences was deleted to make it clear.
Comment # 4: The literature review seems incomplete. Line 140. “The fresh water production unit, whose purpose it is to supply part of the water subscribers' usage, is not utilized by the EH used by these authorities.” While the references [22-23] do not explore water, there is other literature on WEH.
There seems to already be literature on studying Resilience of the Integrated Energy System
https://www.sciencedirect.com/science/article/pii/S1876610216314783
Provides a scenarios-based approach to design WEH. How is the proposed approach of the authors different and better?
https://www.sciencedirect.com/science/article/pii/S0360544220300384
Another paper in the literature demonstrates how to improve resiliency
https://www.sciencedirect.com/science/article/pii/S0378779622008069
This paper specifically considers resiliency against natural disasters:
https://www.sciencedirect.com/science/article/pii/S2352152X22000457
As indicated in the overall comments, the authors are requested to compare their methods against those in the literature.
Authors’ Response: Thank you very much for your comment. Thank you for mentioning. Table 1 is added to the current version of the paper to compare this approach with related references.
Comment # 5: In the introduction, from line 147, the statements of these innovations must be improved to be clear and rigorous:
- ‘Effective structure of MCES’ is vague language. Kindly specify what the effective structure is and how it has resiliency against natural catastrophes.
- The term ‘scenario-based approach’ has been used without any context or prior explanation
- There is existing literature on scenario-based approach. The authors do not include it in the introduction.
- Investigating the resilience allocation of WEH ….. to resiliency improvement of the MGs
- The sentence is unclear and repeats itself with the term resiliency.
- There is existing literature on WEH. The contribution of this paper is unclear.
- Is the “exploration of effective allocation of WEH such that it improves resiliency during a natural disaster” the main contribution?
- It is unclear what the authors mean by "analyzing an economic validation."
Authors’ Response: Thank you very much for your comment.
- After carefully reviewing the related literature, we came up with the idea of multi water and energy (WEH) for resiliency improvement of the networked MGs. For this purpose, different scenarios were considered for assessing the resiliency of the network against natural catastrophic event that causes serious damages on the network by analysing the energy-not-supplied (ENS) factor. Since the MGs are incorporated with renewable energy sources (RESs), scenario-based method is presented to capture the uncertainty associated with the electrical, heat, water demands and electricity, natural gas and water prices. In this work, we assumed that the heat and water system is out caused by the natural catastrophe, and the heat and water consumption of the MG buses is supplied only by the WEHs. This concept has not been followed by related literature.
We have also provided a methodology that shows the structure that is used for improving the resiliency of networked MGs against natural disasters. In addition, Table 1 clearly shows the contributions of this work compared to the related literature.
- More references regarding the scenario-based approach are added to the revised version of the paper and Table I shows the contribution of the work compared to other studies.
- The contributions are clarified and Table 1 is added to compare the contribution of this work compared to other studies.
Comment # 6: As in the first review, it is unclear why two WEH are used. The authors are requested to clarify this point.
Authors’ Response: Thank you very much for your comment. The basic concept of this work to analyse the effectiveness of multi WEH on the networked MGs. To perform such analysis, two WEH units were considered and allocated in the network according to the outcomes of the optimization method. It is worth mentioning that more WEH could be considered in the analysis, but it may not seem to be cost-effective, since the demands are already satisfied and it is more financially relational than cases with more WEHs.
Comment # 7: Please explain in the text what (?) is. Is there a difference between and ? The equation uses whereas Figure 1 uses .
Authors’ Response: Thank you very much for your comment. They are the same and the equation is corrected in the revised version of the paper.
Comment # 8: Line 230: “Increasing and improving the resilience level of the grid is equivalent to increasing the ? index.”
While it is a plausible statement, could the authors provide reference to other work where such indices have been used and shown to improve resilience.
Authors’ Response: Thank you very much for your comment. The equation (1) indicates the definition of resiliency and for instance, reference [1] used this formula to start off their mathematical formulation.
[1] A. Kavousi-Fard, M. Wang, and W. Su. "Stochastic resilient post-hurricane power system recovery based on mobile emergency resources and reconfigurable networked microgrids." IEEE Access 6 (2018): 72311-72326.
Comment # 9: Figure 2 must be better explained:
- The figure must be explained better. For those unfamiliar with using PDF and CDF, this process is unclear. How is the PDF generated? Walking through an example here will greatly help.
- How is scenario reduction done? Is this a formal mathematical process or it is a manual one based on the authors' experience?
- What do the authors mean by scenario aggregation?
Authors’ Response: Thank you very much for your comment. More explanation is added to the section 4 of the revised version of the paper that clarifies the process of scenario-based method. The input parameters of the optimization problem include electrical, heating and water demands as well as electricity, natural gas and water prices. The proposed uncertain parameters are modeled using Gaussian probability distribution function (PDF). For modelling, the standard deviation of the uncertain parameters is 0.15 of the mean of parameters values. Figure 3 represents the flow chart of scenario-based method. By simulating the Roulette Wheel approach (commonly known as the fitness-proportion selection, is a way to randomly select from a given list of weighted inputs.), 1000 scenarios are obtained. The data value for each parameter is uncountable and undefined. Based on the high number of scenarios and them being time-consuming to process, the generated scenarios have been reduced to 10 best possible ones based on the backward reduction algorithm [32]. Reference [32] is added to the revised version of the paper that explicitly explains the steps of scenario-based approach and readers are kindly referred to this reference. Scenario reduction is a mathematical process and can be done by different methods [32].
Also, the scenario aggregation refers to the process of aggregating multiple scenarios to generate more complex scenarios to make the modelling more accurate robust.
Comment # 10: Figure 3: A legend to indicate what the colors represent would be helpful.
Authors’ Response: Thank you very much for your comment. Thank you for mentioning. Each figure shows the generated scenarios based on the scenario-based approach. Since a legend with 10 labels could conceal the figures, it was not added to the figures. However, more explanation is added to the caption of Figure 4 to clarify it.
Comment # 11: Figure 4. The authors are requested to explain the figure or provide a legend for what the various numbers, symbols, and arrows mean.
Authors’ Response: Thank you very much for your comment. A legend is added to the figure in the revised version of the paper for more clarification.
Comment # 12: Line 402: indicated there are 4 scenarios. Earlier, Figure 2 showed the roulette-wheel creates 1000 scenarios, and then downselection to 10. The connection between Figure 2 and this section is unclear. The authors are requested to clarify.
Authors’ Response: Thank you very much for your comment. The 10 scenarios are the output of the scenario-based method, where firstly 1000 scenarios were generated based on the data of the uncertain parameters as inputs of the mathematical modelling. These scenarios represent the worst cases that could occur on the input data of the problem (output generation of the generators, prices, demands, etc.). These parameters make the problem uncertain and need to be captured which is done by the scenario-based method. For assessing the resiliency of the studied structure against natural disaster, critical outages where considered for performance evaluation of the network when subversive damages occur in the network due to these destructive events. Basically, the resiliency assessment is done for each single one of the obtained scenarios that represent the worst cases from the perspective of the input data of the problem (output generation of the generators, prices, demands, etc.).
Comment # 13: Line 428 and line 457: The authors do not prove that the two indicated sites are the best.
Eq. 1 provides a metric for resiliency. The authors are requested to present a formal approach to finding the best sites and the ideal number of WEH using formal optimization approaches where the R index is maximized.
It would be good to see how the R index changes as the location and number of WEH change.
Authors’ Response: Thank you very much for your comment. The basic concept of this work to analyse the effectiveness of multi WEH on the networked MG. To perform such analysis, two WEH units were considered and allocated in the network according to the outcomes of the optimization method. It is worth mentioning that more WEH could be considered in the analysis, but it may not seem to be cost-effective from the perspective of the optimization method, since the demands are already satisfied and it is more financially rational than cases with more WEHs. At each operation of the optimization problem, the solver seeks the optimal location for the WEHs meanwhile it calculates the objective function and attempts to satisfy the constraints. Once all found out, the process would end. Also, we were not seeking to propose a sensitivity analysis by evaluating the alternation of the value of R with regard to the location and number of WEHs and this could be a future work.
Comment # 14: The results shown on Line 488 are not rigorous. The authors start with a case without WEH, and hence there is low resiliency. The introduction of WEH adds energy to the system and thereby the metric the authors have defined has increased.
The authors do not compare their proposed approach to existing methods in the literature to quantify the benefit.
Authors’ Response: Thank you very much for your comment. we would like to thank dear reviewer for their concern regarding the advantages of this work over the related references. To clarify this and present the contributions of this work, Table 1 is provided in the revised version of this work. It is worth mentioning that no previous work studied the effect of multi WEHs on the resiliency of networked MG considering both uncertainties and subversive damages resulted from a catastrophic event. We mainly tried to focus on the advantages of having multi WEH in the networked MG in different scenarios of critical damages and two perspectives of financial cost and resiliency were followed to evaluate the study, rather than how the operation changes when having more than one WEH.
Comment # 15: What is the objective function here? Please provide an equation. Is this different from the resiliency index of Eq. 1?
Authors’ Response: Thank you very much for your comment. Equation (1) is the general definition of the resiliency. This definition can be presented by the expected load recovery from the damage time until the load recovery, or the expected energy not supplied due to the occurred subversive event. These definitions are presented via equations 4 and 5. The objective function is the total energy not supplied cost and the operation cost of the WEHs which include the WEHs internal operation cost. In Tables 3 and 4, the main objective function is in the last column (Eq 6) and its underlying functions are provided as well for more analysis.
Comment # 16: Is this uncertainly analysis the same as the roulette wheel analysis done before? It is unclear how the authors arrived at these numbers.
Authors’ Response: Thank you very much for your comment. The stochastic analysis is done by considering the scenarios obtained from the scenario-based method. This analysis has different outcomes since the uncertain parameters associated with the prices, demands and generation units are also considered in the analysis.
Comment # 17: Investment cost is 21151× in one case and 18286.4 in the other? That is a 10000x difference. This may be a typo.
Authors’ Response: Thank you very much for your comment. We apologize for the typo. It is corrected in the new version of paper.
Comment # 18: 1.4154 e8 and 141333546 are inconsistent. Kindly use scientific notation in both cases
Authors’ Response: Thank you very much for your comment. These numbers are also scientifically noted in the revised version of the paper.
Comment # 19: Typos, grammar, stylistic comments, quality improvements
The paper can still benefit from drastically improving the language to make readability easier. Below is an incomplete list of suggestions:
- The sentence starting on Line 121 is very long. Please consider breaking it into shorter sentences to improve readability.
- Figure 1, 2, and 7 are low quality images and blur when zoomed in. The authors are requested to improve them.
- Line 241: Hence, we can expansion expand the resiliency index as follows
- Line 243: the less energy or supply is not supplied to the network, the higher the index. Please fix the grammar of the italicized words
- 6. Please use large brackets
- There should be a space between the reference type and the reference number. Example: Eq. 6 instead of Eq.6.
Authors’ Response: Thank you very much for your comment. The figures’ quality are increased and all typos and issues are corrected.
Author Response
Reviewer # 3:
Authors’ Response: Firstly, we thank you for your accurate attention and valuable recommendations. We are glad to hear your positive feedback and would like to thank you for giving us constructive suggestions to improve the quality of the paper.
Comment # 1: Line 57: The increasing growth of roads and lack of appropriate equipment in the 57 network are two of the main factors that prolong equipment repair time and recovery of 58 lost loads during a distribution network fault.
Having more roads should reduce the time to access any faults, so should reduce repair time.
Authors’ Response: Thank you very much for your comment. It is corrected in the revised version of the paper.
Comment # 2: Line 72: In recent years, extensive research and investigations have brought a new notion to the 72 subject of energy trading among the negotiators. This sentence does not make sense. What is this new notion? Which negotiators?
Authors’ Response: Thank you very much for your comment. The new notion refers to the idea of having different energy units capable of generating power in the network and providing a platform for negotiating of these units over energy and price. For instance, EHs can connect to the electrical grid or MGs and act as an independent units and sell/buy power to/from the main grid.
Comment # 3: Line 105 gas infrastructures. Natural 105 gas and also water are not mentioned. This does not make sense. Presumable gas infrastructures are used for natural gas. So what was mentioned in [12].
Authors’ Response: Thank you very much for your comment. We apologize for these issues and they are corrected in the revised version of the paper.
Comment # 4: Lines 327-366. Would it not be better to select the scenarios to be used based on historical records and likelihoods of them occurring rather than a random Roulette Wheel approach.
Authors’ Response: Thank you very much for your comment. The scenario-based approach is utilized to capture the uncertainties associated with the uncertain parameters of the input data such as demands, prices and generations. For assessing the resiliency in the even of catastrophic situations, critical scenarios were considered that could cause serious damages to the network and threaten its operation.
Comment # 5: Lines 337 and Figure 3. It is useful to mention, what area (Iran) the load curves are obtained for. Many areas in the world have sufficient solar panels to have a low demand during the day time (Duck Curve). The typical load curves in my area look very different. Also many areas of the world have no heat distribution infrastructure and only distribute water and electricity. It would be more realistic if figure 3 was based on actual demand statistics and the optimisation is then used to match similar demand statistics.
Authors’ Response: Thank you very much for your comment. Thank you for your explanations and kind suggestions. The data used in this paper is drawn from the reference [31], in which the demands of a typical day were considered for the analysis. However, we would consider this idea of real performance evolution of this work for our future works.
Comment # 6: Line 251 objective function is equivalent to the energy not supplied to the network. (so the units should be kWh). In table 3 the Objective Function is shown in $ (141.54 M$). My comment made earlier “Table 2. Figures 5 and 6 show that the power required is > 1 MW. A 1MW gas generator costs about $500k (US). How can that be done with an investment cost of $18286. Since most of the energy required is heat and a heat pump is typically 400% efficiency, is it cheaper to generate this heat using a heat pump?”
Authors’ Response: Thank you very much for your comment. The objective function of the problem is defined in eq. 6 and it is corrected in the new version of the paper. According to the authors in [2], the prices of natural gas is maximum 3 cent/KW. The natural gas as the inputs of the boiler and CHP units is purchased from the natural gas network, the modeling of which was out of scope of this paper. Since the scope of this paper was to analyse the effectiveness of multi WEH on the networked MG, details of the WEH had to be considered for the modeling. Also, the CHP unit has this capability of converting gas to electricity which also supports the electrical demands as well.
[2] Roustai M, Rayati M, Sheikhi A, Ranjbar A. A scenario-based optimization of Smart Energy Hub operation in a stochastic environment using conditional-value-at-risk. Sustainable cities and society. 2018 May 1; 39:309-16.
Comment # 7: Table 2 has an investment cost of $18286.4 and an Objective function of $141.33546M. Does this mean that by spending $18.286k one can save $141.33546M. That seems incredible and should be explained. Also my comment above does not seem to have been addressed in the new version. “Heat Pump” is not mentioned in the text and the $18.286k is still << the $500k cost of a typical 1MW gas generator. Table 3 has an investment cost of $2115M that seems very expensive for supplying 33.8 kW (for a 10 hour period?). Tables 2 and 3 and their description are not clear to me and seem contradictory.
Authors’ Response: Thank you very much for your comment. The total objective function is the total of cost of WEHs operation, investment cost and the cost of supplying the lost loads during the catastrophic event. Some typos were in the table and the table is corrected in the revised version of this paper such as the investment cost with uncertainty and objective function values in Table 4 are scientifically noted.
Comment # 8: I am still very concerned about the relevance and significance of this paper. The justification and aim of the energy hubs need to be explained more. Are they there as buffers to be used all the time. Are they to be there to only be used in the event of natural catastrophes (hurricanes, flood and earthquakes), which occur very rarely. Under those conditions, the power distribution systems will be severely damaged and the power demand will be changed dramatically as a result. The type of power demand will be dramatically different distributing heat as piped hot water or steam will be a very low priority. Electricity distribution will be a high priority. That can be used to pump and purify water, and provide some basic heating for people. During a catastrophe it is unlikely that industrial heating will be used. I can’t see the justification for providing pumped hot water or steam.
Authors’ Response: Thank you very much for your comment. We would like to thank dear reviewer for their concern regarding the contributions of this work. Table 1 is provided to clearly state the contributions of this work compared to some other related references. It is crystal clear the WEHs are not operated only during the disruptive events, since this is not a financially rational idea and such disasters rarely happens. However, it does not mean that the supporting different types of loads that are lost in such disasters is neglectable. The WEHs as independent investors can connect to the MG and support the network in the normal event and the critical event. Also, this contribution of valuing restoration of different types of loads is what this work pursued and no prior work implied it according to the best of our knowledge.
Comment # 9: Having energy hubs to provide resilience against natural catastrophes is worthwhile. The cost benefit is not clearly stated and the costs in table 2 don’t make sense. A 1 MW gas generator costs about $500k (US). Table 2 has an investment cost of $18286.4. That does not make sense.
The authors do some costing, but do not include any depreciation costs, while these hubs remain idle, in-between natural disasters. I can’t see how this can be justified. Having a large grid battery may be more economical since that can provide electrical grid services.
Authors’ Response: Thank you very much for your comment. As we mentioned before, authors in [2] provided the natural gas price which is maximum 3 cent/kw and the natural gas for the operation of CHP and boiler units will be purchased from the gas network, the model of which is beyond the scope of this paper. The scenario-based approach was utilized to capture the uncertainty associated with the uncertain parameters of the input data such as demands, prices and generations. For assessing the resiliency in the even of catastrophic situations, critical scenarios were considered that could cause serious damages to the network and threaten its operation. Also, in this paper, we attempted to value all types of demands and find new ways of supporting lost loads at the same time in the event of disasters.
Comment # 10: I still cant really see why much more heat power is required than electrical power (fig6).
Authors’ Response: Thank you very much for your comment. Authors’ Response: Thank you very much for your comment. We would like to thank dear reviewer for their concern regarding the contributions of this work. Table 1 is provided to clearly state the contributions of this work compared to some other related references. It is crystal clear the WEHs are not operated only during the disruptive events, since this is not a financially rational idea and such disasters rarely happens. However, it does not mean that the supporting different types of loads that are lost in such disasters is neglectable. The WEHs as independent investors can connect to the MG and support the network in the normal event and the critical event. Also, this contribution of valuing restoration of different types of loads is what this work pursued and no prior work implied it according to the best of our knowledge.
Comment # 11: Any simulation should use historical disaster event data, rather than random (roulette) based values.
Authors’ Response: Thank you very much for your comment. Thank you very much for your comment. The scenario-based approach is utilized to capture the uncertainties associated with the uncertain parameters of the input data such as demands, prices and generations. For assessing the resiliency in the even of catastrophic situations, critical scenarios were considered that could cause serious damages to the network and threaten its operation.
Round 3
Reviewer 2 Report
I thank the authors for addressing the concerns in this new manuscript.
I request the authors update the abstract, introduction, and conclusion sections to better reflect the changes made to the other sections. In particular, the scope of the research. The authors expressed this well in their response:
After carefully reviewing the related literature, we came up with the idea of multi water and energy (WEH) for resiliency improvement of the networked MGs. For this purpose, different scenarios were considered for assessing the resiliency of the network against natural catastrophic event that causes serious damages on the network by analysing the energy-not-supplied (ENS) factor. Since the MGs are incorporated with renewable energy sources (RESs), scenario-based method is presented to capture the uncertainty associated with the electrical, heat, water demands and electricity, natural gas and water prices. In this work, we assumed that the heat and water system is out caused by the natural catastrophe, and the heat and water consumption of the MG buses is supplied only by the WEHs. This concept has not been followed by related literature. Also, it should be mentioned that the studied 24-bus networked MGs case is chosen to perform the simulations on it and demonstrate the effectiveness of the work. This approach is applicable on other networked MGs that have the RESs and are able to host WEHs.
A version of the above can be used to update the abstract, introduction, and conclusion.
The authors are requested to improve the quality of the figures
- All figures: text size standardized as per the journal guidelines
- Figure 3: gray shaded boxes with white text are unnecessary. Unshaded boxes with black borders and black text will improve readability
Author Response
I thank the authors for addressing the concerns in this new manuscript.
Authors’ Response: Firstly, we thank you for your accurate attention and valuable recommendations. We are glad to hear your positive feedback and would like to thank you for giving us constructive suggestions to improve the quality of the paper.
Comment # 1: I request the authors update the abstract, introduction, and conclusion sections to better reflect the changes made to the other sections. In particular, the scope of the research. The authors expressed this well in their response:
After carefully reviewing the related literature, we came up with the idea of multi water and energy (WEH) for resiliency improvement of the networked MGs. For this purpose, different scenarios were considered for assessing the resiliency of the network against natural catastrophic event that causes serious damages on the network by analysing the energy-not-supplied (ENS) factor. Since the MGs are incorporated with renewable energy sources (RESs), scenario-based method is presented to capture the uncertainty associated with the electrical, heat, water demands and electricity, natural gas and water prices. In this work, we assumed that the heat and water system is out caused by the natural catastrophe, and the heat and water consumption of the MG buses is supplied only by the WEHs. This concept has not been followed by related literature. Also, it should be mentioned that the studied 24-bus networked MGs case is chosen to perform the simulations on it and demonstrate the effectiveness of the work. This approach is applicable on other networked MGs that have the RESs and are able to host WEHs.
A version of the above can be used to update the abstract, introduction, and conclusion.
Authors’ Response: Thank you very much for your comment. The abstract, introduction and conclusion updated by using the response as mentioned above in the new version of this paper. Please see the paper.
Comment # 2: The authors are requested to improve the quality of the figures
- All figures: text size standardized as per the journal guidelines
Authors’ Response: Thank you very much for your comment. All of the figures were modified and standardized in the new version of this paper, maybe when the Word file is converted to a pdf file by the journal, the quality of the figures is changed. Please see the paper.
Comment # 3: Figure 3: gray shaded boxes with white text are unnecessary. Unshaded boxes with black borders and black text will improve readability
Authors’ Response: Thank you very much for your comment. Figure 3 changed based on your comment in the new version of this paper. Please see the paper.
Reviewer 3 Report
Just 2 philosophical comments:
The authors reply indicates that according to [2] (2021) "the natural gas price which is maximum 3 cent/kw". That figure is out of date, thanks to Russia invading the Ukraine.
https://www.visualcapitalist.com/mapped-global-energy-prices-by-country-in-2022/
shows that the March 2022 world wide average Natural Gas prices is 9c(US)/kWh with double that in much of Europe. That price has fallen since the peak, but the present price in many countries is still >> 3c(US)/kWh. That may effect some of the conclusions. I also notice that the price in Iran is very much lower than anywhere else. That will effect the structure of Energy Hubs.
I notice that the paper states " Natural catastrophes have become more common in recent years44 as a result of climate change and global warming [1]". To minimise its effect, gas consumption needs to reduce and it is likely that a carbon price much in excess of the 3c/kWh will be imposed. The European carbon price is already in excess of that. The paper acknowledges that climate change is a problem, but the proposed gas dominated energy hubs seem to be more like business as usual solution to minimise any inconvenience. Relying more on Solar and Batteries would seem to me a more forward solution. In my mind that detracts from the importance of the paper.
Author Response
Just 2 philosophical comments:
Authors’ Response: Firstly, we thank you for your accurate attention and valuable recommendations. We are glad to hear your positive feedback and would like to thank you for giving us constructive suggestions to improve the quality of the paper.
Comment # 1: The authors reply indicates that according to [2] (2021) "the natural gas price which is maximum 3 cent/kw". That figure is out of date, thanks to Russia invading the Ukraine.
https://www.visualcapitalist.com/mapped-global-energy-prices-by-country-in-2022/ shows that the March 2022 world wide average Natural Gas prices is 9c(US)/kWh with double that in much of Europe. That price has fallen since the peak, but the present price in many countries is still >> 3c(US)/kWh. That may effect some of the conclusions. I also notice that the price in Iran is very much lower than anywhere else. That will effect the structure of Energy Hubs.
Authors’ Response: Thank you very much for your comment. A review of the data of the International Energy Agency shows that in 2020, among the twenty countries in the world with the highest amount of subsidies paid in natural gas, Iran ranks first with the allocation of 12.180 billion dollars to gas subsidies with a share equal to 33%. In the 10-year period from 2010 to 2020, Iran is in the first place in the world by paying 324.471 billion dollars in gas subsidies. This is despite the fact that during this period of time, the amount of electricity subsidy paid in Iran is almost equivalent to the total amount of subsidy paid by 11 countries in the world. However, the amount of gas subsidy payment in Iran during this period of time was almost equal to the total subsidy of 16 countries in the world. Iran's gas fields refer to the natural gas reserves in the hydrocarbon reservoirs of Iran, which is estimated at 1200 trillion cubic feet (33 trillion cubic meters). Iran ranks second in the world in terms of gas reserves and has 17% of the world's gas. Iran is considered to be the second country with the largest gas resources. The existence of huge resources of gas and its cheap price in the country on the one hand and the country's energy policies on the other hand have led to the increasing intensity of natural gas consumption in the country.
Comment # 2: I notice that the paper states " Natural catastrophes have become more common in recent years44 as a result of climate change and global warming [1]". To minimise its effect, gas consumption needs to reduce and it is likely that a carbon price much in excess of the 3c/kWh will be imposed. The European carbon price is already in excess of that. The paper acknowledges that climate change is a problem, but the proposed gas dominated energy hubs seem to be more like business as usual solution to minimise any inconvenience. Relying more on Solar and Batteries would seem to me a more forward solution. In my mind that detracts from the importance of the paper.
Authors’ Response: Thank you very much for your comment. Definitely, considering solar and wind energy helps to reduce the consumption of fossil fuels. But here, due to critical conditions and the need to supply critical loads in a very short period of time, we are forced to use gas fuel in the energy hub. However, supplying critical loads near the grid and considering the electricity and gas energy at the same time increases efficiency and consequently reduces energy wastage.